# Flexible and Transparent Luminescent Cellulose-Transition Metal Cluster Composites

**DOI:** 10.3390/nano13030580

**Published:** 2023-01-31

**Authors:** Maria Amela-Cortes, Noée Dumait, Franck Artzner, Stéphane Cordier, Yann Molard

**Affiliations:** Université de Rennes, CNRS, ISCR–UMR 6226, ScanMAT–UAR 2025, IPR–UMR 6251, F-35000 Rennes, France

**Keywords:** NIR phosphorescence, metal atom clusters, biodegradable polymer

## Abstract

Red-NIR luminescent polymers are principally obtained from petroleum-based derivatives in which emitters, usually a critical raw material such as rare-earth or platinum group metal ions, are embedded. Considering the strong ecological impact of their synthesis and the major risk of fossil fuel energy shortage, there is an urgent need to find alternatives. We describe a luminescent nanocomposite based on red-NIR phosphorescent molybdenum nanoclusters, namely Cs_2_Mo_6_I_8_(OCOC_2_F_5_)_6_, embedded in an eco-friendly cellulose biopolymer matrix that is obtained by a simple solvent casting technique. While homogeneity is kept up to 20 wt% of cluster complex doping, annealing hybrids leads to a large increase of their emission efficiency, as demonstrated by quantum yield measurements.

## 1. Introduction

The Earth’s overall pollution coming from petroleum-derived plastics can be reduced by designing new materials based on low-cost, abundant, and renewable resources. In this frame, eco-friendly biodegradable polymers will play a major role in replacing conventional polymer composites for sustainable development [1,2,3]. Cellulose is one of the most abundant and versatile natural biopolymers [4,5]. Cellulose is biocompatible, biodegradable, and presents a high mechanical strength, which makes it an ideal candidate for applications in biomedicine, optoelectronic, anti-counterfeiting, and integration in flexible devices [6,7,8,9,10,11]. To target optoelectronic applications, emissive dyes should be embedded in the cellulose matrix. However, one drawback on processing cellulose is its high crystallinity, which does not allow dissolution in most solvents, and thus, precludes the incorporation of dyes by solution processes. Using cellulose ester derivatives, such as cellulose triacetate (**CTA**) or cellulose acetate phthalate (**CAP**), is an efficient alternative in this case. Indeed, such derivatives are well-known for their film coating properties [12,13], and appear as appealing matrices to design functional nanocomposites. In particular, photoactive nanocomposites combine the emission properties of inorganic dyes with the flexibility and optical transparency of the cellulose matrix [14,15]. Examples of inorganic dyes incorporated into cellulose or cellulose derivatives are based mainly on critical raw materials, such as rare earth ions [16,17,18,19] and semiconductor nanoparticles based on CdSe/Zn [20,21,22], CuInS_2_ [23], or more recently, ZnO [24]. Octahedral transition metal clusters compounds (MC) of formula A*_n_*[M_6_Q_8_^i^L_6_^a^] (A = alkali; M = metal atom; Q = inner ligand, halogen or chalcogen; L^a^ = apical ligand, halogen, organic ligands, Figure 1) are an emerging class of luminophores [25]. In particular, MC containing molybdenum, an abundant, non-toxic, and cheap element compared to noble or rare earth metals, are highly emissive in the red NIR region when excited in the UV-Vis range [26,27]. Recently, we showed that such emission is also observed when MC are excited in the NIR via a two-photon absorption process [28]. They are obtained by high temperature synthesis as crystalline powders exhibiting a strong ceramic-like behavior. Powder chemical compositions can be further modified to afford highly soluble compounds. Molecular compounds built-up from cluster units are phosphorescent with emission lifetimes in the range of several tenths of a microsecond and show a considerable Stokes shift that minimizes reabsorption. Depending on Q^i^ and L^a^ ligands, MC quantum yield can be adjusted from a few percent to nearly unity [29,30,31,32]. MC have been functionalized to be integrated in liquid crystals or polymers for applications in anti-counterfeiting [33], solar concentrators [34], wave-guiding materials [35,36], inkjet printing [37,38], or lightening [39,40]. As efficient singlet oxygen generators, they have also been studied as an emissive probe for biological oriented applications, such as antibacterial surfaces [41], in photodynamic therapy against various cancer cells [42,43,44,45,46] or as oxygen sensors [47,48,49,50]. Hence, MC integration into an ecofriendly host matrix represents a new step in the design of sustainable materials containing such emitter. However, light scattering, which is primarily due to physical segregation between the inorganic and organic components, is an issue that one must overcome when dealing with hybrid composites for optoelectronic devices. That is the reason why most of the previously cited luminophores were embedded at low content into their organic host. Here, we show that CAP is suitable as polymer matrix to support the luminescent pentafluoropropionate octahedral molybdenum cluster complex Cs_2_Mo_6_I_8_(OCOC_2_F_5_)_6_ **(MoIP)** for optoelectronic devices. Transparent, robust films were obtained for concentrations of **MoIP** up to 20 wt%. The effect of MC content embedded in the **CAP** matrix, on the thermal and the optical properties of hybrid nanocomposites were investigated.

## 2. Materials and Methods

Silverpentafluoropropionate was purchased from Aldrich (Aldrich, Darmstadt, Germany, 98%, CAS 509-09-01). Cellulose acetate phthalate (**CAP**) was purchased from Aldrich (Aldrich, Mw = 2534.12 g/mol, CAS 9004-38-0). The compound was obtained starting from Cs_2_Mo_6_I_14_ and silverpentafluoropropionate following published procedures [40]. In brief, a solution of silver pentafluoropropionate (0.935 g, 3.42 mmol) in 10 mL of acetone was added to a 20 mL solution of Cs_2_Mo_6_I_14_ (1.5 g, 0.52 mmol) in acetone, under argon and in the dark. The mixture was stirred for 48 h in the dark and filtered through a Celite^®^ pad. A red-orange powder was obtained after complete evaporation. ^19^F-NMR (acetone-d_6_): *δ* (ppm) = −83 (3F), −120 (2F). EDAX: Cs 2, Mo 8, I 11, F 77, no Ag.

*Cluster complex@CAP mixtures and film deposition.* Amounts of cluster complex and **CAP** were calculated to obtain 2 g of composite sample. A first solution containing 1, 10, 20, or 30 wt% of cluster complex dissolved in 5 mL of HPLC grade acetone was prepared by mixing the right amount of cluster with acetone and stirring for 5 min at 25 °C. A second one was obtained by dissolving **CAP** in 20 mL of HPLC grade acetone and heating at 50 °C for 30 min under stirring. Once all **CAP** was dissolved, the corresponding solution of the cluster complex was added to the CAP containing solution and the mixture was stirred for 1 h at 50 °C. Finally, the mixtures were poured into glass petri dishes and the solvent was slowly evaporated. Films of 0.25 mm thickness were obtained.

*FTIR spectroscopy.***CAP**, **MoIP** cluster complex, and composite films were all analyzed by infrared spectroscopy using Universal Attenuated Total Reflectance Accessory FT-IR Spectroscopy (Bruker, vertex 70, Billerica, MA, USA). Spectra were recorded in the range from 500 to 5000 cm^−1^ and analyzed using OPUS 7 software (Bruker, Billerica, MA, USA).

*Thermal analysis.* DSC measurements were carried out in a DSC 25 differential scanning calorimeter (TA Instruments, New Castle, DE, USA), calibrated using purified indium (99.9%) as the standard reference material. Samples (3–4 mg) were cut off and placed in an aluminum pan. They were heated at a constant rate of 10 K min^−1^ using dry atmosphere of argon as carrier gas, in a temperature range of 30 to 200 °C. Thermograms were analyzed by TRIOS 5.4 Software (Waters, TA Instruments, New Castle, DE, USA).

Thermogravimetric analyses were performed at 10 K min^−1^ on a TGA/DT Perkin Pyris Diamond. (Perkin Elmer, Waltham, MA, USA).

*Small Angle X-ray scattering (SAXS).* X-ray diffraction patterns were collected with Pilatus 300k detector (Dectris, Baden, Switzerland) mounted on home-made Guinier setup based on a Cu K_α_ microsource (λ = 1.541 Å) from Xenocs (Grenoble, France). The sample to detector distance (277 mm) has been calibrated by using silver behenate. The X-ray patterns were therefore recorded for a range of reciprocal spacing q = 4πsinθ/λ from 0.01–1.75 Å^−1^ where θ is the diffraction angle. The experiments performed with the present set-up provide accurate measurements of distances between 600 Å and 3.8 Å. The acquisition time was 1 h. Film samples were placed perpendicularly to the beam. The scattering intensities as a function of the radial wave vector were determined by circular integration.

*Photophysical properties.* Transmission measurements were recorded on a Perking Elmer Lambda 35 UV-vis spectrophotometer. Lifetime measurements and TRPL mapping were realized using a picosecond laser diode (Jobin Yvon deltadiode, 375 nm, Horiba France SAS, Palaiseau, France) and a Hamamatsu C10910-25 streak camera mounted with a slow single sweep unit (Hamamatsu Photonics France, Massy, France). Signals were integrated on a 30 nm bandwidth. Fits were obtained using origin software and the goodness of fit judge by the reduced *χ*^2^ value and residual plot shape. The luminescence spectra in deaerated dichloromethane and absolute quantum yields in the solid state were measured with a C9920–03 Hamamatsu system equipped with a 150 W xenonlamp, a monochromator, an integrating sphere, and a red-NIR sensitive PMA-12 detector.

## 3. Results and Discussion

Homogeneous flexible films with 1, 10, 20, and 30 wt% of MC labelled **MoIPx@CAP** (with x = 1, 10, 20, or 30, respectively), were prepared from mixtures of precursor solutions by a simple solvent casting method, as described in the Section 2.

Infrared spectroscopy was used to observe the influence of metal cluster integration on the organic backbone and *vice versa*. Figure 2 compares IR spectra recorded between 750 and 2000 cm^−1^ for neat **CAP**, **MoIP,** and nanocomposites. The most intense bands for both neat compounds correspond to the C=O stretching vibrations and is located at 1720 cm^−1^ for **CAP** and 1670 cm^−1^ for **MoIP**. Modifying the cluster content in **CAP** modifies the intensity ratio between these two bands with no apparent shift (highlighted in blue in Figure 2). Characteristic bands of **CAP**, at 1034 and 743 cm^−1^ corresponding to the C-O-C stretching of the pyranose ring and to the o-phenyl, respectively, are observable in all films and do not seem to be modified upon cluster integration.

The most obvious effect is observed for the **CAP** bands located at 1598 and 1580 cm^−1^ corresponding to the C=C stretching of the conjugated aromatic ring of the phtalate moities that merges in only one absorption band located at 1587 cm^−1^ when more than 20 wt% of cluster is integrated (highlighted in yellow in Figure 2). For **MoIP**, interactions with the host matrix induce a shift of several absorption bands located at 1388 cm^−1^, 1322 cm^−1^, and 1026 cm^−1^ that correspond to the stretching of C-F and C-C bonds within the CF_2_-CF_3_ groups. Such bands shift to 1402 cm^−1^, 1303 cm^−1^, and 974 cm^−1^, respectively (highlighted in red in Figure 2) [51]. These shifts indicate that the structural cohesion and organization within the nanocomposites involve weak interactions between **CAP** and **MoIP** leading to a high homogeneity of the hybrids at low cluster concentration, as depicted in Appendix A.

The small angle X-ray scattering (SAXS, Figure 3) technique was used to determine the influence of cluster integration on the polymer chains packing. The absence of sharp signals in all SAXS scattering patterns reflects the amorphous character of the nanocomposite and indicates a uniform distribution of MC. All scattering patterns of **CAP** and **MoIPx@CAP** exhibit a correlation peak around 5 nm^−1^ corresponding to a CAP inter-chain distance of about 1.18 nm. Increasing **MoIP** content slightly shifts this peak from q = 5.3 nm^−1^ for **CAP** up to q = 4.9 nm^−1^ for **MoIP30@CAP**. This broad X-ray scattering, corresponding to an interdistance of about 1.3 nm, is due to the host matrix and confirms its amorphous character. In the meantime, a low q signal, whose position maximum decreases and whose intensity increases, appears upon increasing the cluster complex content. This broad scattering is well fitted by a liquid organization of the clusters with a mean interdistance lying between 4.3 nm and 7.5 nm. Indeed, the cluster diameter being about 1.2 nm, such large interdistance can only be exhibited when clusters show a very good dispersion. The X-ray scattering intensity increase at very small angles could find origin in cluster aggregation processes but only at high concentration.

The influence of cluster concentration on the thermal properties of the host **CAP** matrix was investigated by differential scanning calorimetry (DSC) and thermogravimetric analysis (TGA). Glass transition temperatures (*T*_g_) and degradation temperatures (*T*_d_) values are gathered in Table 1 (Appendix A for DSC thermograms). The *T*_g_ observed for **CAP** film is 157 °C, which is lower than that described in the literature of around 171 °C. As described by Sakellariou et al. and Roxin et al., the presence of residual solvent from the film casting process, usually acetone, can affect greatly the *T*_g_ [37,52]. This is in good accordance with TGA analysis, which shows, for all samples, a weight loss of around 3% ending at 100 °C for CAP and around 125 °C for doped samples (Figure 3b). However, we wish to point out that *T*_g_ values were measured during the second cooling cycle, when samples have already been heated twice up to 200 °C. Therefore, no traces of residual solvent are expected after such thermal treatment. For doped films, the *T*_g_ decreased from 153 °C for **MoIP1@CAP** to 141 °C for **MoIP30@CAP**. This plasticizing effect is attributed to the introduction of the bulky cluster core and the mobile pentafluoropropionate apical ligands that can disrupt the organization of **CAP** chains in a glassy state, as already observed for PMMA doped with the same cluster unit [40]. For all samples, two major decomposition steps were evidenced by thermogravimetric investigations. The first step starts at around 150 °C and the second one depends on the **MoIP** content within the nanocomposite. For neat **CAP**, all the acetyl and phtalyl groups are lost between 150 and 300 °C, and the backbone chain decomposes between 325 and 400 °C to afford only a carbon residue [4,53]. The first degradation step occurs at lower temperatures for doped CAP samples: 270 °C for **MoIP1@CAP** and 250 °C for **MoIP10@CAP**, **MoIP20@CAP,** and **MoIP30@CAP**. This slight decrease is imparted to the degradation of cluster compound apical ligands that occurs around 250 °C [39,40]. The final decomposition temperature decreases accordingly from 374 °C for neat **CAP** to 297 °C for **MoIP30@CAP**. The residual weight at 400 °C increases, as expected, from neat **CAP**, 12%, as the inorganic content increases to 36% for **MoIP30@CAP** (Table 1).

The introduction of **MoIP** in the **CAP** matrix leads to a composite exhibiting specific absorption and emission properties. Transparent films were obtained for concentrations of **MoIP** up to 20 wt% (see Appendix A and Figure 4). Figure 4a presents transmittance spectra recorded for all samples while Figure 4b,c show the corresponding films deposited on illustrations to show their transparency, as well as the bending ability of **MoIP1@CAP**. Neat **CAP** film is colorless and highly transparent with a transmittance of 90% over the whole range of visible light. The hybrid nanocomposite films are transparent and colored. **MoIP1@CAP** is light yellow showing a transmittance of 89% at 600 nm and a cut-off at around 430 nm. Cut-off at the edge of visible spectrum (400–430 nm) has been reported for halogenated Mo cluster complexes **(TBA)_2_Mo_6_Cl_14_** embedded in low content (approx. 0.1 wt%) in acrylate blends and corresponds to the absorption of cluster complex [34]. The coloration of films evolves to deep orange for **MoIP10@CAP** and **MoIP20@CAP** that present both a similar behavior showing a transmittance of 85% at 600 nm and a cut-off at 550 nm. For **MoIP30@CAP**, the sample loses its transparency due to light scattering caused by phase segregation. Hence, the weak interactions between the **CAP** and **MoIP** allow up to 20 wt% of cluster complex content without segregation. At this point, we wish to emphasize that such doping concentration in hybrid films, obtained by simple mixing and solvent casting, is already high and demonstrate the potential of weak interactions in the integration of inorganic components in organic matrices [54,55].

When excited with UV-blue light, **MoIP** emits efficiently in the red NIR with an emission centered at 650 nm and an absolute quantum yield (Φ_em_) in the solid state of 0.35 [29,30,31,39,40,50]. As shown in Table 2, the **MoIP** quantum yield value (Φ_em_) is poorly affected by the embedment in the **CAP** matrix. Interestingly, Φ_em_ does not decrease with the increasing concentration of the cluster complex. Actually, particle–particle interactions can lead to non-radiative excited state and thus to a quenching of the emission [34,39,40]. It is also well known that a modulation of the O_2_ concentration deeply affects the cluster emission efficiency [47,48,50]. Indeed, as MC are phosphorescent, their excited triplet state is efficiently quenched by triplet oxygen, thus generating singlet oxygen, which is of particular interest for applications such as oxygen sensing, photodynamic therapy of cancer, and antibacterial treatment [42,43,47,48,50,56]. Therefore, the emission efficiency of **MoIPx@CAP** is essentially linked to the oxygen concentration within the host matrix that is directly correlated to its gas permeability and to the power of the excitation beam [33]. Absolute quantum yield measurements, performed alternatively under air and N_2_ atmosphere, demonstrate that at low MC content, hybrids are less sensitive to oxygen. Indeed, before thermal annealing (*vide infra*), passing from an air to a N_2_ saturated atmosphere leads to an AQY increase from 0.29 to 0.34 for **MoIP1@CAP**, while for **MoIP30@CAP** the AQY value increases from 0.35 to 0.42. Hence, increasing **MoIP** content leads to a higher oxygen sensitivity, which is assessed by the MC plasticizing effect on the host matrix. As observed by SAXS experiments, increasing the cluster content induces an increase of the polymer inter-chains distance, thus improving its gas permeability.

Emission decay profiles recorded after excitation at 375 nm were fitted with one or two components depending on the cluster concentration (see Appendix A for emission decay maps, profiles, and fits). For homogeneous samples, a monoexponential behavior is observed, which shows, on one hand, that the cluster complex integrity is preserved in the hybrid (no apical ligand exchange occurs with the residual hydroxyl groups contained in **CAP**) and, on the other hand, that MC behave as molecular discrete species like in solution which is in good accordance with SAXS observations. **MoIPx@CAP**; x =1, 10, show a monoexponential behavior with calculated lifetime values close to 120 µs, indicating that **MoIP** complexes are well protected from oxygen quenching when embedded in **CAP** matrix. In contrast, the **MoIP** precursor lifetime value in deaerated acetone is 269 µs, while it drops to 2 µs under air [57]. For **MoIP20@CAP** and **MoIP30@CAP**, the appearance of two components may arise from isolated clusters and segregated ones. We further investigated the hybrids photostability towards constant irradiation for 60 min at 25 °C and thermal treatment after 30 min of annealing at 180 °C. Obtained results are gathered in Figure 5b,c for **MoIP10@CAP**. We observed that the emission intensity of **MoIP1@CAP** or **MoIP10@CAP** increases by 175% or 125%, and reaches a plateau after 10 or 5 min, respectively, during continuous irradiation under our experimental conditions. This phenomenon is already observed when the same cluster anion is embedded in PMMA or in liquid crystal matrices, which is attributed to the time necessary to reach the equilibrium between the surrounding triplet oxygen consumption and the matrix gas permeability [33,39,40,58]. Temperature dependent photostability was investigated after heating the samples to 180 °C (well above their glass transition temperature) and cooling to 30 °C at a rate of 10 K min^−1^. Surprisingly, the emission intensity at 30 °C is largely enhanced after a first heating and cooling cycle, by 135, 210, and 125% for composites doped at 1, 10, and 20 wt%, respectively (see ESI for temperature dependent photostability studies, Appendix A). These results are consistent with the AQY values calculated before and after annealing (Table 2) at a low cluster content. For **MoIP20@CAP**, the difference is not observable in Φ_em_ values, and on the contrary, **MoIP30@CAP** and the neat powdered **MoIP** complex show a loss of their emission intensity of about 25 and 20%, respectively, after annealing at 180 °C. Hence, when **MoIP** is homogeneously embedded in **CAP**, it becomes more emissive after annealing. This might be attributed to some macromolecular reorganizations and to the loss of residual solvent molecules coming from the casting process, as suggested by TGA, leading to a better packing of the polymer chains that induces a lower gas permeability [59,60]. The cluster complex plasticizing effect is then minimized until the cluster concentration becomes too high to be compensated by the polymer chains packing. To the best of our knowledge, this is the first time that such phenomenon is observed in cluster-doped hybrid systems.

## 4. Conclusions

In conclusion, this work presents highly red-NIR phosphorescent hybrid nanocomposite materials containing stable molybdenum cluster inorganic dyes embedded within an eco-friendly cellulose polymer. Homogeneous, transparent, and stable films are obtained by a simple mixing of two solutions containing the individual component, followed by drop casting and evaporation. As demonstrated by IR studies, the structural cohesion between the cluster complex and the **CAP** matrix involves only weak interactions. Indeed, the native bands corresponding to the stretching of carbonyl functions of the host and inorganic moiety are poorly affected within the hybrid and only the phtalate moieties of **CAP** seem affected by the MC integration.

SAXS experiments reveal that MC are in a liquid-like organization within **CAP**, meaning that they are homogeneously distributed within the host matrix. Such assumption was further confirmed by time-correlated emission studies for samples containing 1 and 10 wt% of MC, which showed an emission decay that could be fitted with only one component. However, nanocomposites lose their transparency for a MC content of 30 wt%, which clearly indicates some phase segregation at such high concentration. The DSC and TGA analysis show that the glass transition temperature, together with the decomposition temperature of the hybrid nanocomposites, decrease with the increase in MC content. These phenomena are assessed by the MC plasticizing effect, which disrupts the organization of the host chains and lowers the thermal stability of MC compared to the **CAP** host, respectively. Emission studies show that clusters behave mostly like discrete species within the nanocomposite with high absolute quantum yield values. A prolongated irradiation induces a large increase of the emission intensity that is inversely proportional to the MC content. Such increase is ascribed to the time needed to reach, under continuous irradiation, a stationary regime between the O_2_ consumption and its diffusion within the host. These already excellent emission abilities can be further improved by thermal annealing, which allows better packing of the polymer chains and the removal of residual solvent traces, in particular for a low MC content. This work opens wide perspectives in the design of eco-friendly bio-based devices with bright red-NIR emission.

## Figures and Tables

**Figure 1 nanomaterials-13-00580-f001:**
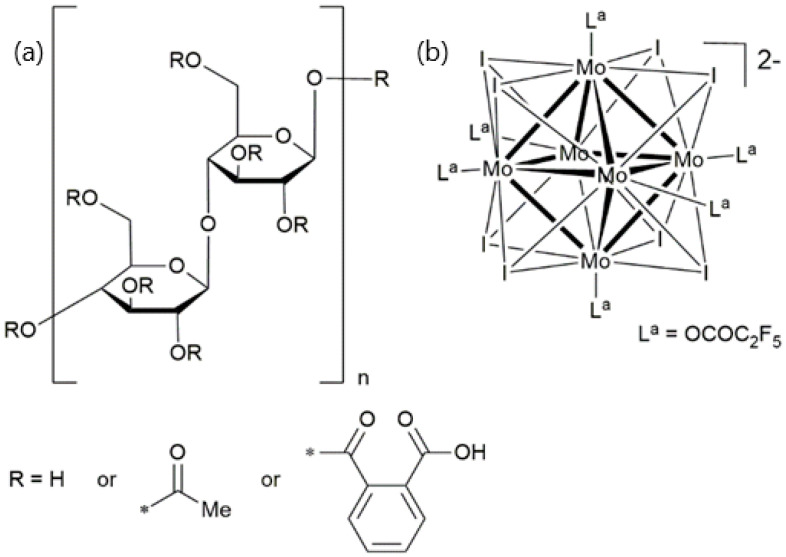
(**a**) Structure of cellulose acetate phthalate. (**b**) Structure of the Mo_6_ metal nanocluster anion used in this work whose anionic charge is counterbalanced by two Cs+ cations. The symbol “*” represents the linking point of the R group.

**Figure 2 nanomaterials-13-00580-f002:**
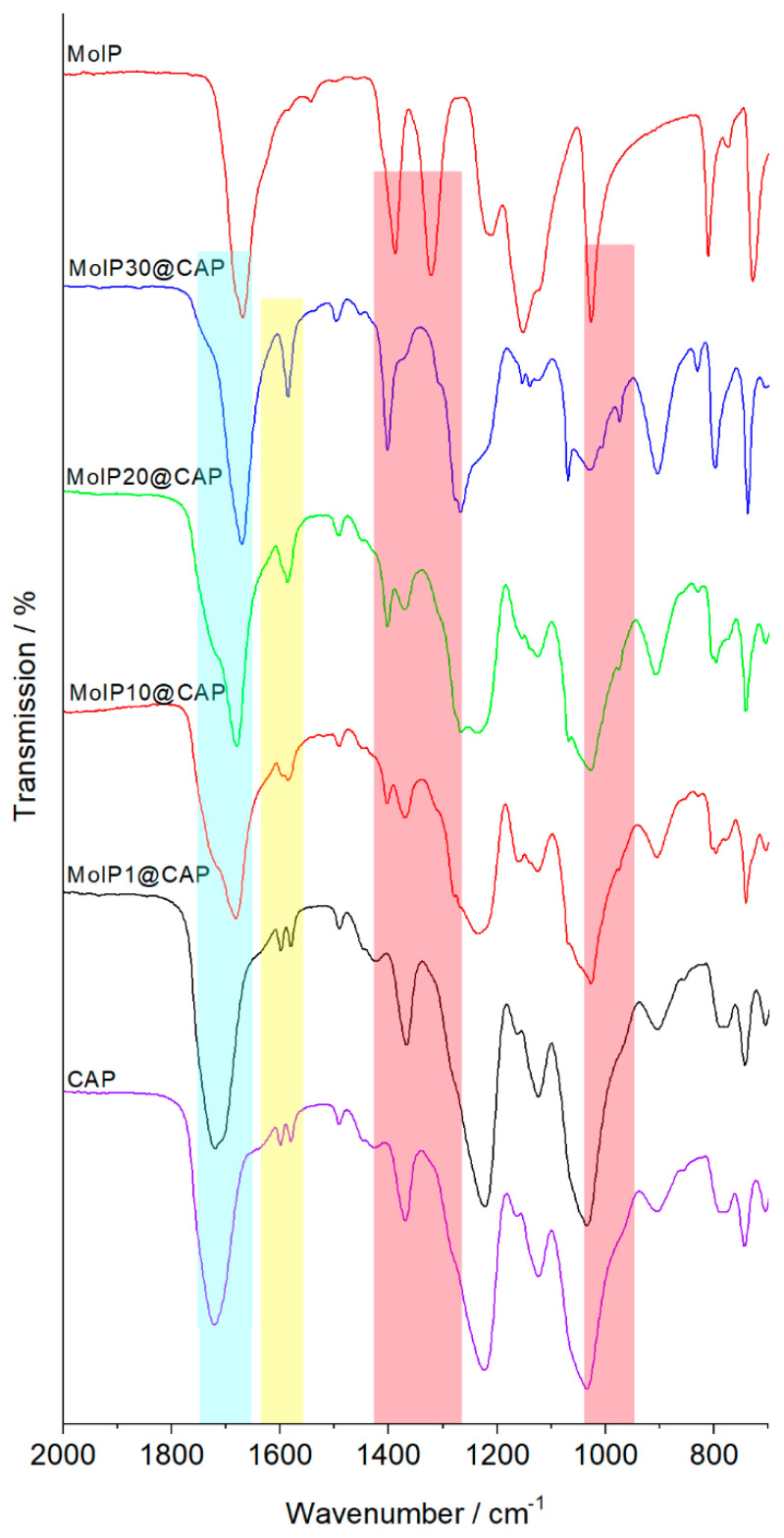
Infrared spectra of neat **CAP**, neat **MoIP,** and hybrid composite **MoIPx@CAP** films. The highlighted bands are the ones discussed.

**Figure 3 nanomaterials-13-00580-f003:**
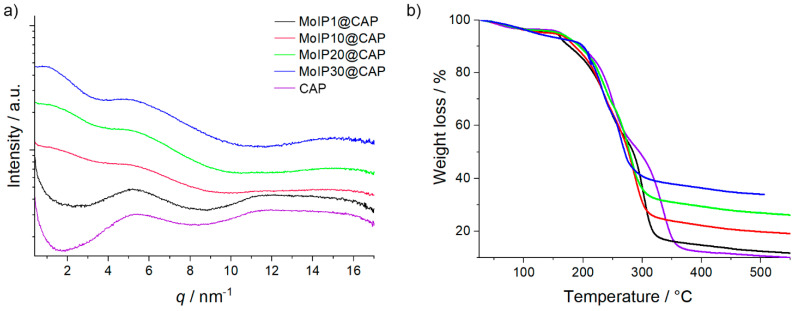
(**a**) Small angle X-ray scattering diffraction patterns at 25 °C. (**b**) Thermogravimetric curves recorded for **CAP** (purple), **MoIP1@CAP** (black), **MoIP10@CAP** (red), **MoIP20@CAP** (green), and **MoIP30@CAP** (blue).

**Figure 4 nanomaterials-13-00580-f004:**
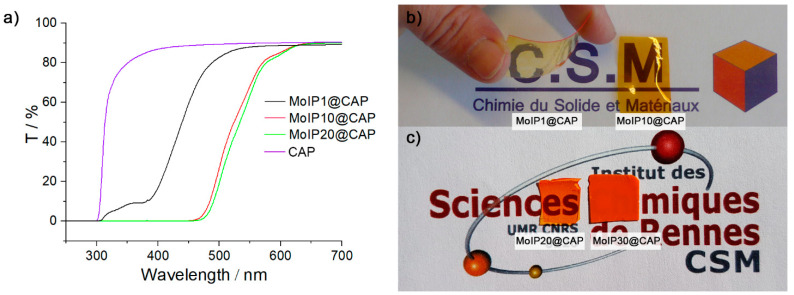
(**a**) Films transmittance of **CAP** (purple), **MoIP1@CAP** (black), **MoIP10@CAP** (red), and **MoIP20@CAP** (green). Pictures and natural light of (**b**) **MoIP1@CAP** and **MoIP10@CAP**, (**c**) **MoIP20@CAP** and **MoIP30@CAP**.

**Figure 5 nanomaterials-13-00580-f005:**
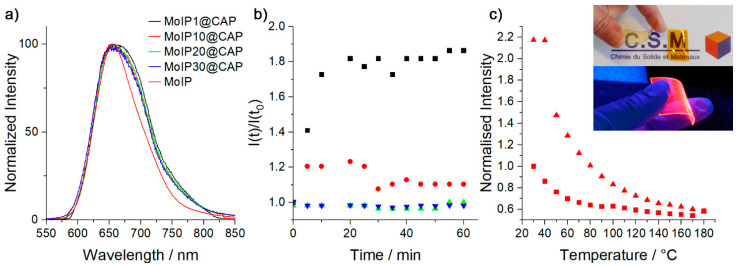
(**a**) Emission spectra of **MoIP1@CAP** (black), **MoIP10@CAP** (red), **MoIP20@CAP** (green), **MoIP30@CAP** (blue), and **MoIP** cluster complex (dark red). (**b**) Photostability under constant irradiation of **MoIP1@CAP** (black square), **MoIP10@CAP** (red circle), **MoIP20@CAP** (green triangle), and **MoIP30@CAP** (blue inverted triangle). (**c**) Photostability studies with temperature for **MoIP10@CAP** (square: on first heating, triangle: on cooling); inset: **MoIP1@CAP** and **MoIP10@CAP** under natural light (top) and **MoIP1@CAP** under UV-A irradiation (bottom).

**Table 1 nanomaterials-13-00580-t001:** Composition and thermal behavior of doped CAP films: *T*_d_: final decomposition temperature, *T*_g_: glass transition temperature determined from the second cooling cycle.

Sample	MC (wt%)	*T*_d_ (°C)	Residue 400 °C (wt%)	Calculated MC Residue (wt%)	*T*_g_ (°C)
**CAP**	0	374	12	0	157
**MoIP1@CAP**	1	324	14	2	153
**MoIP10@CAP**	10	310	22	10	153
**MoIP20@CAP**	20	305	30	18	141
**MoIP30@CAP**	30	297	36	24	141

**Table 2 nanomaterials-13-00580-t002:** Emission properties of the cluster complex and the doped CAP films.

Sample	Φ_em_ ^a^		λ_max/_nm	FWHM/cm^−1^	τ ^d^/μs (%)	τ_av_ ^e^
	air	N_2_				
**MoIP**	0.35	-	650	1920	5 (28) 50 (72)	48
**MoIP1@CAP**	29 ^b^/45 ^c^	34 ^b^/47 ^c^	653	2180	113	113
**MoIP10@CAP**	24 ^b^/34 ^c^	32 ^b^/40 ^c^	658	2128	113	113
**MoIP20@CAP**	23 ^b^/19 ^c^	32 ^b^/27 ^c^	650	2128	144 (41) 99 (59)	122
**MoIP30@CAP**	35 ^b^/27 ^c^	42 ^b^/36 ^c^	657	2156	123 (66) 67 (33)	111

^a^ λ_exc_= 365 nm; ^b^ before and ^c^ after annealing at 180 °C; ^d^ λ_exc_ = 375 nm; ^e^ average lifetime is calculated as τ = (Σa_i_τ_i_^2^)/(Σa_i_τ_i_).

## Data Availability

Not applicable.

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
