# Peer review of "Flexible and Transparent Luminescent Cellulose-Transition Metal Cluster Composites"

_nanomaterials, 2023, doi:10.3390/nano13030580_

Round 1

Reviewer 1 Report

The present paper deals with the investigation of photoluminescent cluster complex synthesis and deposition on the cellulose substrate. The paper is interesting and well-written. The topic is actual and perspective to broad contribution to the sci-tech in the near future.

However, there are several issues and suggestions to be solved before publishing.

1) The cluster complex Cs2Mo6I8(OCOCF2CF5)6 synthesis process should be briefly discussed.

2) line 72: Is cluster complex really "soluble" in acetone?  Maybe "dispersed". Therm 'nanocomposite' is used in the text.

3) How "solution" was prepared? mixing conditions? Please add details.

4) Figures 4 b and c, should identify the shown samples.

5) Can plasticization be obtained by incomplete drying of acetone from films? Please discuss it in relation to TGA data.

6) Please describe the agglomeration of clusters in the nanocomposites. What is the critical threshold concentration ?

Author Response

Rev1: The cluster complex Cs2Mo6I8(OCOCF2CF5)6 synthesis process should be briefly discussed.

Us: The synthesis of this compound is described since 2014 in ref 40. A brief recall has been added to the materials and method section.

Rev1: line 72: Is cluster complex really "soluble" in acetone?  Maybe "dispersed". Therm 'nanocomposite' is used in the text.

Us: Cluster compounds are species lying between molecules and nanoparticles. Their molecular character allows them to be soluble in various solvents (usually polar solvent when their cation is alkali) such as acetone or ethanol. Therefore, they are readily soluble in various solvents whose nature depends on the lipophilicity of their cation. In our case, the 6 perfluorinated chains in apical position allows to increase the solubility of the cluster complex into many solvents, including dichloromethane and chloroform.

Rev1: How "solution" was prepared? mixing conditions? Please add details.

Us: Solutions were prepared by dissolution of the mentioned compounds and stirring. Few words have been added to assess so and give more information in the materials and method section.

Rev1: Figures 4 b and c, should identify the shown samples.

Us: Samples are identified in the legend of figure 4b and 4c. In order  to ease the reading, we modified all graphs by using the same color for each sample and by integrating the legend within the graph. We are confident that now graphs are easier to follow.

Rev1: Can plasticization be obtained by incomplete drying of acetone from films? Please discuss it in relation to TGA data.

Us: We thanks the reviewer for this comment. The TGA data show a first slight step between 50 and 100 °C (around 3 wt% loss), which could be attributed to acetone but also to moisture. TGA curves have been added to the text in Figure 3.

Tg values were measured by DSC and taken from the second cooling cycle, i.e.  samples have been heated up to 200°C twice before measurements. Moreover, first and second cooling cycles are perfectly superimposable. Hence, we expect that no trace of solvent remain in the samples when the Tg values are taken. The fact that the Tg decreases as the amount of cluster increases is therefore independent of the amount of solvent. We have already observed this behavior by embedding the same pentafluoropropionate cluster into PMMA matrix.(ref 40).

Some sentences have been added in the text to give more insight to the reader. We also added some more explanation related to the presence of residual solvent in relation with the emission behavior before and after annealing. We hope Reviewer 1 will find these changes satisfactory.

Rev1: Please describe the agglomeration of clusters in the nanocomposites. What is the critical threshold concentration ?

Us: self-aggregation of inorganic phases in a organic phase always occur when interactions between both components are not strong enough. In particular for high inorganic concentrations.  It is very difficult to assess about a critical threshold concentration. In our case, we rely on the transparency of the obtained films in the visible region. We observed that at 20 wt% of doping, samples were transparent which shows that aggregation did not reach the minimum size for the light to be scattered in the visible range. It is clear that at 30 wt% such limit is reached. Hence, most probably the threshold concentration lies between these two values.

Reviewer 2 Report

In this paper, the authors have prepared and characterized transition metal composites with flexible and transparent luminescence. The paper has been written well. However, certain concerns need to be noticed and clarified:

1.     In the introduction, the results of other scholars' research need to be added to compare with those done by the author to highlight the strengths and innovations of the author's research.

2.     The format of the “MOIP30@CAP” statement in Table 1 is not consistent with the format of the other statements, please correct.

3.     The author highlights the flexibility of composites in the title, which should be one of the more important advantages of composites, but this aspect is not highlighted too much in the article, please add a characterization of this section.

4.     The data graphs of the representations suggest adding legends to make them easier for the reader to read.

5.     There are some formatting issues in the manuscript, so please check the manuscript carefully for errors and correct them.

6.     The conclusion needs to be strengthened by summarizing the results obtained and highlighting the advantages of the paper's methodology.

Author Response

Rev2: In the introduction, the results of other scholars' research need to be added to compare with those done by the author to highlight the strengths and innovations of the author's research.

Us: Results from other scholars have been added in the introduction and in the ms. 11 references have been added to the text. Ref : 11, 14, 15, 19, 22, 24, 41, 43, 44, 45 and 46.

Rev2: The format of the “MOIP30@CAP” statement in Table 1 is not consistent with the format of the other statements, please correct.

Us: We thank the Reviewer 2 for this comment. This has been corrected.

Rev2: The author highlights the flexibility of composites in the title, which should be one of the more important advantages of composites, but this aspect is not highlighted too much in the article, please add a characterization of this section.

Us: Reviewer is right. We highlight the flexibility of the composite which is in large contrast with the ceramic character of the native cluster powder. In fact, by using this term, we wish to emphasize that, as in all 3D nanomaterials, our nanocomposite combines the physical properties of the host matrix and the functionalities of the integrated nanocomponent: the emission properties of the metal cluster.

The flexibility, in fact the bending ability, is highlighted in the TOC, in figure 4b and in the inset of figure 5c where film samples are held and bended between two fingers. Moreover, cellulose is known for its flexibility as reported by various authors like in:

An, N. L.; Qin, J. X.; Zhou, X.; Wang, Q. D.; Fang, C. Q.; Guo, J. P.; Nan, B. Recent Progress in Cellulose-Based Flexible Sensors. J. Renew. Mater. 2022, 10 (9), 2319-2334. DOI: 10.32604/jrm.2022.021030.

Zhao, D. W.; Zhu, Y.; Cheng, W. K.; Chen, W. S.; Wu, Y. Q.; Yu, H. P. Cellulose-Based Flexible Functional Materials for Emerging Intelligent Electronics. Adv. Mater. 2021, 33 (28). DOI: 10.1002/adma.202000619

Pang, B.; Jiang, G. Y.; Zhou, J. H.; Zhu, Y.; Cheng, W. K.; Zhao, D. W.; Wang, K. J.; Xu, G. W.; Yu, H. P. Molecular-Scale Design of Cellulose-Based Functional Materials for Flexible Electronic Devices. Adv. Elec. Mater. 2021, 7 (2). DOI: 10.1002/aelm.202000944.

Chen, Z. Y.; Yan, T.; Pan, Z. J. Review of flexible strain sensors based on cellulose composites for multi-faceted applications. Cellulose 2021, 28 (2), 615-645. DOI: 10.1007/s10570-020-03543-6.

Purandare, S.; Gomez, E. F.; Steckl, A. J. High brightness phosphorescent organic light emitting diodes on transparent and flexible cellulose films. Nanotechnology 2014, 25 (9). DOI: 10.1088/0957-4484/25/9/094012.

Min, S. H.; Kim, C. K.; Moon, D. G. Flexible Top Emission Organic Light Emitting Diodes with Ni and Au Anodes Deposited on a Cellulose Paper Substrate. Mol. Cryst.  Liq. Cryst. 2013, 584 (1), 27-36. DOI: 10.1080/15421406.2013.849422.

Min, S. H.; Kim, C. K.; Lee, H. N.; Moon, D. G. An OLED Using Cellulose Paper as a Flexible Substrate. Mol. Cryst.  Liq. Cryst. 2012, 563, 159-165. DOI: 10.1080/15421406.2012.689153.

Two of these references have been added in the introduction to support the flexible character of cellulose films.

Rev2: The data graphs of the representations suggest adding legends to make them easier for the reader to read.

Us: To ease the reader, we modified all figures and integrated the legend in all of them. Moreover, all presented data respect the same color code. We hope that the manuscript will be easier to follow.

Rev2: There are some formatting issues in the manuscript, so please check the manuscript carefully for errors and correct them.

Us: We tried to correct as much formatting issues as possible.

Rev2: The conclusion needs to be strengthened by summarizing the results obtained and highlighting the advantages of the paper's methodology.

Us: The conclusion has been rewritten. We hope that Reviewer 2 finds it appropriate.

Additional corrections: value of FWHM in table 2 are given in cm-1 and not in nm.

Round 2

Reviewer 2 Report

Accept as it is.